# Exploring the Potential of *Muridae* as Sentinels for Human and Zoonotic Viruses

**DOI:** 10.3390/v16071041

**Published:** 2024-06-27

**Authors:** Ilaria Di Bartolo, Luca De Sabato, Giovanni Ianiro, Gabriele Vaccari, Filippo Maria Dini, Fabio Ostanello, Marina Monini

**Affiliations:** 1Department of Food Safety, Nutrition and Veterinary Public Health, Istituto Superiore di Sanità, Viale Regina Elena, 299, 00161 Rome, Italy; ilaria.dibartolo@iss.it (I.D.B.); luca.desabato@iss.it (L.D.S.); giovanni.ianiro@iss.it (G.I.); gabriele.vaccari@iss.it (G.V.); marina.monini@iss.it (M.M.); 2Department of Veterinary Medical Sciences, University of Bologna, Via Tolara di Sopra, 50, Ozzano dell’Emilia, 40064 Bologna, Italy; filippomaria.dini@unibo.it

**Keywords:** *Muridae*, rats, mice, viral zoonoses, rodents, reservoir, Italy

## Abstract

In recent years, the transmission of viruses from wildlife to humans has raised significant public health concerns, exemplified by the COVID-19 pandemic caused by the betacoronavirus SARS-CoV-2. Human activities play a substantial role in increasing the risk of zoonotic virus transmission from wildlife to humans. Rats and mice are prevalent in urban environments and may act as reservoirs for various pathogens. This study aimed to evaluate the presence of zoonotic viruses in wild rats and mice in both urban and rural areas, focusing on well-known zoonotic viruses such as betacoronavirus, hantavirus, arenavirus, kobuvirus, and monkeypox virus, along with other viruses occasionally detected in rats and mice, including rotavirus, norovirus, and astrovirus, which are known to infect humans at a high rate. A total of 128 animals were captured, including 70 brown rats (*Rattus norvegicus*), 45 black rats (*Rattus rattus*), and 13 house mice (*Mus musculus*), and feces, lung, and liver were collected. Among brown rats, one fecal sample tested positive for astrovirus RNA. Nucleotide sequencing revealed high sequence similarity to both human and rat astrovirus, suggesting co-presence of these viruses in the feces. Murine kobuvirus (MuKV) was detected in fecal samples from both black (n = 7) and brown (n = 6) rats, primarily from urban areas, as confirmed by sequence analysis. These findings highlight the importance of surveillance and research to understand and mitigate the risks associated with the potential transmission of pathogens by rodents.

## 1. Introduction

Over the past few years, the transmission of viruses from wildlife hosts has led to severe diseases affecting humans, like the recent COVID-19 pandemic caused by the betacoronavirus SARS-CoV-2, which is believed to have originated from wildlife and subsequently spread to humans [1], causing a widespread global pandemic with significant impacts on public health, economies, and societies worldwide.

Human activities, including increasing population density, global travel, changes in territory utilization, and urban expansion, have intensified interactions between humans and animals, thereby significantly contributing to the spread and transmission of zoonotic viruses to humans [2]. Furthermore, other factors such as the increase in immunocompromised and transplant populations, global warming, and decreased vaccination rates may also play a role in the emergence and re-emergence of zoonotic viral diseases [3].

The largest group of mammals found worldwide belong to the *Rodentia* order, specifically within the *Muridae* family [4], which represents the primary source of zoonotic infectious diseases for humans [5,6,7].

*Muridae* populate urban and peri-urban areas in large numbers, often residing in close proximity to humans compared to other wild animals.

As described for SARS-CoV-2 [8], rats can serve as hosts for human viruses, some of which may replicate in various organs. In other cases, rats may serve as accidental hosts, as observed with human viruses such as NoV or hepatitis E virus genotype 3 (HEV-3), which have been detected in rats but without definitive evidence of replication [9,10]. However, this finding raises concerns about the potential for reverse-zoonotic transmission of emerging variants back to rodent species, including wild rat populations [10].

Brown rats (*Rattus norvegicus*), black rats (*Rattus rattus*), and house mice (*Mus musculus*), members of the order *Rodentia*, family *Muridae*, are synanthropic rodents widely distributed in urban, peri-urban, and rural environments, where they live in close proximity to humans and livestock. Urban rats, being hosts to numerous zoonotic pathogens, could potentially serve as reservoirs for these pathogens, playing a pivotal role in the maintenance, transmission, and occasional spillover of infectious agents [5,7].

However, there is limited information available on the factors facilitating the transmission of viruses carried by rats or mice that may pose a zoonotic threat to human health. Viruses can spread among rats through direct or indirect contact with excrement or saliva. Given that urban brown rats live closely with humans and have access to food sources, transmission from rats to humans is likely to happen through direct contact, inhalation of aerosols containing rat excrements or saliva, or via bites [11,12].

Rodents can host many different viruses, either zoonotic or not, from various families including *Adenoviridae*, *Arenaviridae*, *Coronaviridae*, *Flaviviridae*, *Hantaviridae*, *Hepeviridae*, *Herpesviridae*, *Paramyxoviridae*, *Parvoviridae*, *Picornaviridae*, *Pneumoviridae*, *Polyomaviridae*, *Poxviridae*, *Reoviridae,* and, more recently, *Hepeviridae* Rat-HEV, belonging to *Orthohepevirus C* species, has been also identified in human cases in several European countries [13,14], although its zoonotic potential is still not clarified. Overall, this confirms the role of rats as hosts to a large plethora of viruses. These viruses can be species-specific, zoonotic, or acquired through reverse zoonoses from other hosts, including humans.

Orthohantaviruses, family *Hantaviridae*, are commonly zoonotic, with *Muridae*, especially rodents, serving as the primary reservoirs. These viruses are mainly transmitted to humans, causing severe diseases, through the inhalation of aerosolized rodent urine, saliva, and feces. This mode of transmission is believed to be prevalent among reservoir host animals [15]. In Europe, four pathogenic hantaviruses have been identified, including Puumala virus (PUUV), Dobrava virus (DOBV), Saaremaa virus (SAAV), and Seoul virus (SEOV), which frequently coexist [16,17]. In 2020, in Europe, 1647 cases of hantavirus infection (0.4 cases per 100,000 population), mainly caused by Puumala virus [18], were reported. Seoul virus (SEOV) has been detected in wild brown rats in several European countries [19,20,21,22]. A recent study in the Czech Republic revealed a high prevalence of orthohantavirus RNA in rodent hosts, with 24.2% (37/153) of animals testing positive [23]. Severe clinical cases of SEOV infections have been documented in the UK and France, with pet rats identified as the source of infection [24,25]. In the UK, confirmation of this transmission route comes from an 8% seroprevalence among farmers exposed to wild rats [26]. In Italy, limited studies have investigated the circulation of arenaviruses and hantaviruses in rodents, uncovering antibody prevalence rates of 0.4% for Puumala virus (PUUV) and 0.2% for Dobrava virus (DOBV) [27]. A recent study conducted in the Turin area (Northwestern Italy) confirmed the detection of hantavirus RNA from different organs in 6 of 41 *Mus domesticus* samples collected [28].

*Arenaviridae* are a family of zoonotic viruses that can be transmitted to humans through contact with rodents or their excreta. They include the lymphocytic choriomeningitis virus (LCMV) and the highly pathogenic Lassa virus (LASV). Symptoms of infection can vary widely, from mild influenza-like symptoms to severe hemorrhagic fever, which can be potentially fatal [29]. The common house mouse, *Mus musculus*, has been considered the primary rodent host for LCMV [27]. Reported LCMV prevalence rates in *Mus musculus* in Europe (from studies in Germany and Spain) have ranged from 3.6% to 11.7% [27].

The monkeypox virus (MPXV) is a zoonotic virus belonging to the *Poxviridae* family, causing an infection that has been reappeared slowly among humans. Since early May 2022, cases of MPXV have been reported in various countries, including Italy, where the disease is not endemic. Consequently, the World Health Organization has declared MPXV a global public health emergency [30]. The virus is transmitted to humans through direct contact with infected animals such as monkeys, rodents, and other small mammals or their body fluids, as well as through human-to-human contact. The source of infection has not yet been definitively proven but rodent species are suspected to be the natural animal reservoir for MPXV [31].

*Muridae* have been identified as a host for nonzoonotic betacoronaviruses (beta-Covs) such as rat coronavirus (RCov), mouse hepatitis virus (MHV), China Rattus coronavirus HKU24 (ChRCoV HKU24), and Longquan Rl rat coronavirus (LRLV) [32]. However, there is evidence suggesting that certain human endemic coronaviruses (OC43 and NL63) may have originated from a rodent reservoir [33]. In vitro and in vivo experiments have demonstrated that some SARS-CoV-2 variants of concern (VOCs) are capable of infecting laboratory rats [34] and that urban rats were exposed to the virus, as proven by anti-SARS-CoV-2 antibodies detection.

Kobuviruses (KoVs) are small, nonenveloped viruses belonging to the *Picornaviridae* family, which infect mammals, including mice, rats, and other small mammals. The infections are often associated with gastroenteritis and transmission occurs via the fecal–oral route [35]. The murine kobuvirus 1 was found to be genetically related to the human Aichi virus, both belonging to Aichi virus A genotype [35], along with feline and canine infecting viruses [36]. KoVs have demonstrated the ability to cross species barriers, as evidenced by interspecies transmission between bovine and porcine KoVs, suggesting that close contact among different animal species could enhance the potential for virus interspecies transmission [37].

Group A rotaviruses (RVAs), *Reoviridae* family, are segmented double-stranded RNA virus and are a common cause of gastroenteritis in their respective host species, particularly in young or immunocompromised animals. Animal rotaviruses typically exhibit species specificity but, due to the segmented nature of their RNA, different genotypes have emerged in the human population through zoonotic transmission or gene reassortment [38]. Earlier studies have documented cross-species transmission between pigs and rats on a pig farm in Brazil [39], as well as between rats and humans in China [40]. However, published literature on RVA in small mammals remains limited.

Noroviruses (NoVs) are single-stranded positive-sense RNA viruses belonging to the *Caliciviridae* family. They are the leading viral cause of human gastroenteritis globally [41]. NoVs have been identified in various animal species, including bovine, porcine, and mice. However, the typical NoVs infecting humans are classified within GI and GII genogroups [41]. Conversely, murine NoVs belong to GV and do not infect humans. Recent studies have detected norovirus RNA belonging to the human GI and GII genotypes in fecal samples of rodents [42,43], suggesting that rats living in an urban area can serve as carriers of human NoVs.

Astroviruses (AstV) belong to the family *Astroviridae*, including the genera Mamastrovirus, which infect mammals; zoonotic infections are rarely documented [44,45]. Human AstV RNA has been previously detected in rat fecal samples and in different sections of the intestine [46]. Interestingly, phylogenetic analyses on genomes have described that rat and the human MLB1 astroviruses may have a common ancestor [47].

Given the increasing proximity of rodents to urban areas in contact with humans and considering that rodents have already been described as reservoirs of zoonotic viruses, we aimed to investigate a plethora of viruses in rats and mice from a North–Central area of Italy. Well-known viruses with established zoonotic potential such as betacoronavirus, hantavirus, arenavirus, as well as the monkeypox virus were included in this study. Additionally, the occurrence of other viruses (such as rotavirus, norovirus, kobuvirus, and astrovirus) belonging to genotypes or serotypes infecting humans, although they have also sporadically been found in wild rats, were also investigated.

## 2. Materials and Methods

### 2.1. Study Areas and Sampling

Animals were captured using mechanical traps between 2020 and 2023 from rural and urban areas across 27 locations spanning five different provinces in two regions in North–Central Italy (area 4.156 km^2^, Figure 1) as part of pest control programs. Data recorded include sex, species (based on external morphology), and body weight. Based on body weight and sex, animals were than assigned to one of two age categories: subadult or adult.

### 2.2. Preparation of Samples

Necropsies were performed using sterile instruments and, depending on the conditions of the carcasses, 85 livers, 103 lungs, and 111 intestinal contents were collected; all samples were frozen and stored at −80 °C until analysis. RNA was extracted from 200 µL of 10% fecal suspension in sterile water (*w*/*v*) obtained by intestinal contents by commercial extraction kits (RNeasy mini kit and Qiamp Viral RNA mini kit, Qiagen, Hilden, Germany). For livers and lungs, RNA was extracted from 25 mg of tissue after homogenization in lysis buffer available in the kit, using the tissue lyser (Qiagen, Hilden, Germany) [48].

### 2.3. Viral Nucleic Acid Detection

Real-time RT-PCR was used for the detection of Sarbecoviruses (betaCoV), RVs, and NoVs GI–GII. Samples were tested by the QuantiFast^®^ Pathogen PCR + IC Kit (Qiagen, Hilden, Germany), as described in previous studies (Table 1).

Conventional end-point RT and nested PCR were used for coronavirus, kobuvirus, astrovirus, arenavirus, and hantavirus by using the Qiagen One-Step RT-PCR Kit (Qiagen, Hilden, Germany) for the first round of RT-PCR and the GoTaq^®^ G2 DNA Polymerase (Promega, Madison, WI, USA) for the second round of nested PCR where applicable, following protocols previously described (Table 1).

Concerning AstV, a nested RT-PCR designed to detect the capsid region of “classic” human astroviruses was employed (Table 1). Subsequently, sequencing was performed using the pan-astrovirus primers pair SF0073 and SF0076, targeting the RdRp (ORF1b gene) (Table 1).

For monkeypox virus, reactions (Table 1) were performed by using the Platinum™ SuperFi™ DNA Polymerase (Thermo Fisher Scientific, Frederick, MD, USA).

The different viruses investigated were not searched in all three different types of specimens. Instead, based on previous studies, each virus was investigated in the sample type where its detection was most likely, either in the organ of replication (e.g., hantavirus or CoV in liver and lung, respectively) or where the virus can accumulate (KoV in feces). In detail, betacoronavirus were investigated in all three types of samples. The detection of rotavirus, norovirus, astrovirus, arenavirus, kobuvirus, and monkey pox nucleic acids were conducted in fecal samples only. Hantavirus RNA was searched for in lung and liver specimens.

### 2.4. Sanger Sequencing

PCR amplicons of expected size were verified by Sanger sequencing via a custom sequencing service (Eurofins Genomics, Ebersberg, Germany). The nucleotide sequences were edited and aligned using the Aliview version 1.28 free software [63]. The related sequences were searched using the BLASTn server on the NCBI GenBank database (http://www.ncbi.nlm.nih.gov/genbank/index.html, accessed on 15 May 2024). Nucleotide sequences were submitted to NCBI with the following accession numbers: PP740880 and PP740881 for astroviruses and PP756626–PP756632 for kobuviruses.

### 2.5. Statistical Analysis

Pearson Chi-square test was applied to investigate the prevalence of MukV and AstV between species, age class (subadult and adult), sex, and place of capture (urban/rural).

Statistical analyses were performed using the software SPSS 28.0 (IBM SPSS Statistics, Armonk, NY, USA) and *p* < 0.05 was set as statistically significant.

## 3. Results

### 3.1. Characteristics of Examined Animals

A total of 128 wild *Muridae* living freely were captured in urban (61.7%) and rural (38.3%) areas and identified, based on morphological features, as follows: 70 brown rats (*Rattus norvegicus*), 45 black rats (*Rattus rattus*), and 13 house mice (*Mus musculus*). Among the examined animals, 43 (33.6%) were females and 85 (66.4%) were males (Table 2). Only adult and subadult animals were retrieved (Table 2).

Due to technical reasons, it was not possible to obtain all three types of biological samples (fecal or rectal swab, lung, and liver) from all the 128 animals. Specifically, 85 livers, 103 lungs, and 111 intestinal contents were collected (Table 2). The three paired specimen types were obtained for 47 animals.

### 3.2. Virus Detection

The samples analyzed from 13 *Mus musculus* (11 lungs, 8 feces, and 6 livers) did not show any detectable RNA of investigated viruses.

Among the 63 fecal samples of *Rattus norvegicus*, one yielded a positive result for AstV RNA. This rat was a subadult male sampled in an urban area. None of the other 17 rats belonging to the same species sampled from the same area during the same period tested positive for AstV. No statistically significant difference (*p* > 0.05) in AstV prevalence was observed by species, sex, age class, and place of capture.

Overall, 13 fecal samples out of 111 tested positive for MuKV. Specifically, 6 out of 40 samples were collected from *Rattus rattus* and 7 out of 63 samples from *Rattus norvegicus*, 10 of which were sampled in the same urban area. Lung and liver samples from the 13 rats that tested positive for kobuvirus in their feces were also assayed and only a lung sample from an adult male brown rat collected in an urban area was also positive for MuKV RNA.


The prevalence of kobuvirus in fecal samples was significantly higher (*p* < 0.05) in subadults (20.5%) than in adults (6.0%) and in urban (19.4%) than in rural areas (2.0%).

Stratifying animals by age class and by the area of trapping, urban rats showed a higher prevalence across both age groups, with the difference being statistically significant (*p* < 0.05) only among adults. No statistically significant difference (*p* > 0.05) in MuKV prevalence was observed by species and sex.

Hantavirus and coronavirus RNA were not detected in 59 lung tissue and 33 liver samples from brown rats, nor in 54 lung tissue and 25 liver samples from black rats.

Additionally, none of the 63 and 40 fecal samples from black and brown rats, respectively, tested positive for RVA, NoV, arenavirus, monkeypox virus, or coronavirus.

Multiple assays (Table 3) were conducted for coronavirus, using both a SARS-CoV-2 and Sarbecovirus real-time and a broad-range RT-PCR. Neither lung, liver, nor feces tested positive for any CoVs.

### 3.3. Sanger Sequencing Confirmation of Positive RNA

According to BlastN alignment, the MuKV strain sequences (no. 6; PP756626–PP756632) belonged to Aichi virus A MuKV and showed the highest nucleotide sequence identities (94–95%) with previously published MuKV sequences identified in rats in China (MW292482–MN646801). MuKV strains displayed high similarities to each other, with nucleotide sequence identities ranging between 94.4 and 100%. MuKV sequences retrieved from feces and lung (PP756626–PP756627) of the same animal were identical, suggesting that the same viral strain was present in both organs. Two out of the six sequences were retrieved from *Rattus norvegicus* and four from *Rattus rattus.* No species-specific virus sequence was observed since the same MuKV sequences were observed in both species.

The positive amplicon for feces of rat52, a subadult male brown rat captured in an urban area, corresponding to the ORF2 genome fragment (Acc. no PP740880) matched with human Astrovirus 1 strains previously detected in human cases and in wastewater across EU countries, including Italy, displaying a nucleotide identity of approximately 99% with the latter. For deeper characterization, the RNA of rat52 was further investigated by amplifying and sequencing the RdRp fragment (PP740881), which matched with typical rat astrovirus strains that could belong to cluster B [64], displaying a nt. id. > 98%.

Despite several attempts to obtain the whole fragment, encompassing ORF1b to ORF2, no amplicon was obtained, leaving open the question of whether the obtained sequences derived from two co-infecting strains or from a new recombinant one and leaving the provisional classification within the cluster unconfirmed.

## 4. Discussion

With the aim of investigating the potential role of rodents as hosts of zoonotic or human viruses, we examined a group of *Muridae* including black and brown rats and mice, captured from urban and rural areas in North–Central Italy. Rural areas were in proximity to small villages or groups of houses, never far from them. The area of sampling was also chosen to investigate animals living in proximity with humans and their garbage or sewage. The study included widely recognized viruses with confirmed zoonotic potential, such as coronavirus, hantavirus, arenavirus, and kobuvirus, as well as the monkeypox virus. Furthermore, it examined the potential occurrence of other viruses (such as rotavirus, norovirus, and astrovirus) belonging to genotypes or serotypes known to infect humans, which have been previously identified in wild rats and mice, although with limited evidence supporting their zoonotic potential [40,42,47].

In our study, no positive results were obtained for the investigated zoonotic viruses, i.e., hantavirus, arenavirus, monkeypox virus, and coronavirus, likely due to the low number of animals analyzed. Although hantavirus is responsible for causing rare zoonosis in South-Eastern European countries, its incidence has been slightly increasing in the last years [65]. Thus, the surveillance of animal hosts in Italy should be continuously performed. Italy shares its border with several countries where hantaviruses, such as PUUV and DOBV, circulate and it is hypothesized also that niches of both European Hantaviruses and their hosts may be found in the Alpe Adria region [66].

Arenaviruses are spread by the same species of rodents that also harbor hantaviruses [27]. Until 2023, the sole arenavirus identified in Europe was LCMV, a “neglected” rodent-borne viral zoonosis [67]. The prevalence of antibodies against the virus in humans varies but may reach 47% in populations with exposure to rodents [68]. In two studies from Northern Italy on the prevalence of LCMV in wild rodents, 5.6% and 6.8% of the tested animals were found to be seropositive [27,69].

Although the reservoir of MPXV has yet to be definitively clarified, evidence suggests that mammals, such as wild rodents, may serve as reservoirs of MPXV. Recently, there has been an increase in human cases of orthopoxvirus (OPXV), including in Italy [70], and sporadic cases of novel OPXVs, which have their reservoir in rodents, have been reported in the last decades [71]. MPXV DNA has also been identified in wastewaters in Europe [52,72,73], raising the concern that small mammals, including rats, living in sewerage systems could potentially become accidental host for MPXV.

No CoVs, either of human or animal origin, were detected in our study. The interest in SARS-CoV-2 was particularly high during the COVID-19 pandemic, driven by efforts to identify antibodies against the virus, and a few reports of viral RNA detection in urban rats were produced [8]. This detection likely resulted from reverse zoonotic transmission, posing a risk of generating new viral variants capable of infecting humans. Studies conducted in England and in Belgium found evidence of antibodies recognizing SARS-CoV-2, with some evidence of neutralization ability suggesting rats were exposed to SARS-CoV-2 [74,75]. However, no evidence of SARS-CoV-2 was revealed in this study. Information regarding the circulation of rodent CoVs in Italy is scarce, with partial data obtained during a virological investigation conducted in 2022, where rodent coronaviruses were identified in mice through sequencing [76]. Besides SARS-CoV-2, the murine coronavirus, also belonging to betacoronavirus, causes a high fatality rate, thus far excluding its detection in the 13 mice investigated in this study. Conversely, rat CoV, despite the high morbidity, is rarely fatal [77].

Astroviruses can infect a wide range of hosts and were initially believed to be highly species-specific, resulting in a classification based on the host species. However, in recent decades, various AstVs have been identified in species not previously reported, including rats, indicating a lack of species barrier. Some rat astrovirus strains were detected in *Rattus norvegicus* in Hong Kong clustering separately from well-known species-specific rat AstV and probably sharing a common ancestor with human MLB1 and MLB-2 AstVs, as evidenced by their genome sequence correlation [47].

In this study, we investigated the presence of human AstV in the feces of investigated animals. Among the 63 samples from brown rats, one tested positive (1.6%), while none of the 40 *Rattus rattus* or the 8 mice showed any positive results. In Italy, there is a lack of prior data on human AstV in rats, despite its known role in causing gastroenteritis [78] and its frequent detection in wastewater [79], confirming its circulation among humans. In our study, sequencing of capsid fragment confirmed the presence of human AstV in the rat feces. Interestingly, the same sample also tested positive for rat AstV, as confirmed by sequencing an RdRp fragment using a different set of primers (Table 1). We presumed but it was not definitively confirmed that both viruses were simultaneously present in the feces. Niendorf and colleagues [46] recently reported the detection of human AstV infection in a rat in Germany, identifying a mixed infection involving a human AstV from genogroup HAstV-8 and a rat AstV, both found in various organs [46]. Since it was proved that AstV can cross intestinal barriers and be detected in spleen [80] and other organs [81], we extended our testing to include the other samples from the AstV-positive rat. However, neither the lung nor the liver tested positive, suggesting that these organs were not currently involved in viral replication in the investigated rat.

HuNoV GI genotype has been detected in brown rats in Denmark [43], while GI and GII genotypes were identified in rats sampled in Hungary and Germany [46] with a similar detection rate (8%) as observed in a study conducted on rodent fecal samples in Finland [42]. The detection of human NoV strains in rats is rare and likely occurs in areas such as wastewater or dump sites [42] where rats come into contact with human feces positive for NoVs. While it is possible for NoVs to be detected in rats living in urban areas, this may not be as common due to limited contact with human feces.

An increasing number of reports have documented the transmission of group A rotaviruses (RVA) between species, from animals to humans and/or between animals [40,82,83,84]. Previous data indicate that rodents could serve as a significant reservoir for rotaviruses, with the potential for these viruses to transmit across species to humans [40,84]. The generation of diversity within the RVA population is influenced by both reassortment and interspecies transmission of RVA strains. As previously described in Italy, in a rat, the genome of RVA identified was closely related to those of RVA from rodents and to the human Wa-like VP6 gene, suggesting interspecies reassortment [83].

MuKV has been detected in samples taken from the lungs, brain, heart, and liver of rodents [85]. Similarly, canine kobuvirus was found in various organs including the brain, lungs, tonsils, and liver of the same puppy [86], suggesting that kobuviruses may exhibit a wide range of tissue tropism. MuKV was previously isolated in Norvegicus rats in Hungary and in pet rats in Italy [87,88]. Our results demonstrate that the strain of MuKV circulating in rats of different species is nearly identical and, as previously observed, can be detected in different organs (in our study, feces and lung), confirming a wide tissue tropism [89]. However, its detection in the lungs could be attributed to the virus circulating in the bloodstream, given that the lung is a highly vascularized organ in rats [90].

Interestingly, rats that tested positive for kobuvirus in our study were predominantly found in the same urban area and among adult animals. Sequencing of positive amplicons revealed only host-specific kobuvirus, typically matching with other murine kobuvirus, confirming that the detected strains were related, suggesting a cluster of infected rats inhabiting the same area and facilitating the spread of kobuvirus through the release of their infected feces.

Our study highlighted a low prevalence or absence of the investigated viruses among Italian rats and mice. It is possible that data obtained reflect limited transmission of such pathogens in this specific population or geographic area. It is known that the geographical distribution and prevalence of zoonotic viruses can vary significantly based on various environmental, ecological, and climatic factors [91]. Other viruses with human as the main host, such as NoV and RVA, may have been absent due to the limited sampling of rodents in urban areas and the predominant focus on peri-urban regions where human fecal contamination is likely infrequent.

The surveillance of synanthropic murids populations for the circulation of viral pathogens within the human population holds significant promise as a tool for anticipating and preparing for future outbreaks. However, further investigation is essential to evaluate the practicality and effectiveness of employing urban rats and mice as sentinels for monitoring viral pathogen circulation. Moreover, future research should also consider the analysis of viruses present in both human and murids populations, facilitating a comprehensive understanding of cross-species transmission dynamics and potential zoonotic threats.

Results obtained are also interesting from the perspective of a One-Health approach; the health of humans and animals is linked to the well-being of the ecosystems in which they reside. Animal diseases, which can be transmitted to humans, represent a public health concern. It has been estimated that around 60% of pathogens causing human diseases have their origins in either domestic animals or wildlife, and 75% of emerging infectious diseases in humans have an animal origin [92].

## Figures and Tables

**Figure 1 viruses-16-01041-f001:**
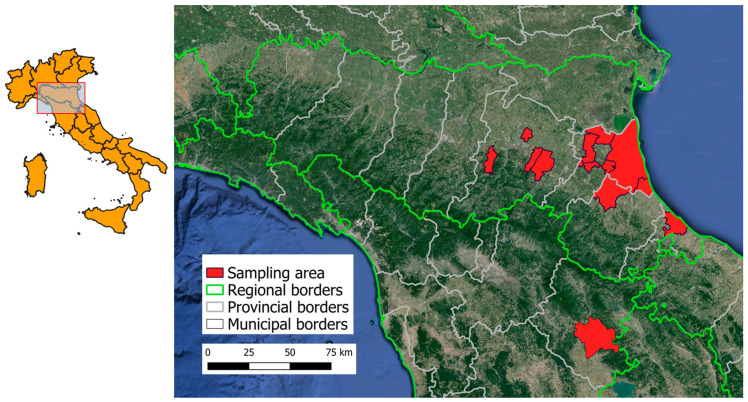
Geographical map showing sampling locations and environment characteristics.

**Table 1 viruses-16-01041-t001:** Methodology description and list of the primers and probe sequences employed for the detection and sequencing of the investigated viruses.

Virus Family	Target Virus/Taxon	Assay	Primers/Probes (5′-3′ Sequence)	References
*Coronaviridae*	SARS-CoV-2	RT-qPCR	RdRp_SARSr-F: GTGARATGGTCATGTGTGGCGG RdRp_SARSr-P2: FAM-CAGGTGGAACCTCATCAGGAGATGC-BBQ RdRp_SARSr-R: CARATGTTAAASACACTATTAGCATA	[49]
	Sarbecoviruses	RT-qPCR	E_Sarbeco_F: ACAGGTACGTTAATAGTTAATAGCGT E_Sarbeco_P1: FAM-ACACTAGCCATCCTTACTGCGCTTCG-BBQ E_Sarbeco_R: ATATTGCAGCAGTACGCACACA	[49]
	Coronavirus	RT-PCR	Pan_CoV_F-1: GGTGGGAYTAYCCHAARTGYGA Pan_CoV_R-1: CCRTCATCAGAHARWATCAT Pan_CoV_R-2: CCRTCATCACTHARWATCAT	[50]
		Semi-nested PCR	Pan_CoV_F-2: GAYTAYCCHAARTGTGAYAGA Pan_CoV_F-3: GAYTAYCCHAARTGTGAYMGH Pan_CoV_R-1: CCRTCATCAGAHARWATCAT Pan_CoV_R-2: CCRTCATCACTHARWATCAT	[50]
*Hantaviridae*	Hantaviruses	RT-PCR, nested	HAN-L-F1: ATGTAYGTBAGTGCWGATGC HAN-L-R1: AACCADTCWGTYCCRTCATC HAN-L-F2: TGCWGATGCHACIAARTGGTC HAN-L-R2: GCRTCRTCWGARTGRTGDGCAA	[51]
*Poxviridae*	Monkeypox virus	PCR, nested	G2R-1st cycle F: ATAGCACCACATGCACCATC G2R-1st cycle R: AAAGGTATCCGAACCACACG MPVX G F mod: GGAAAGTGTAAAGACAACGAATACAG MPVX G R mod: GCTATCACATAATCTGAAAGCGTA	[52]
*Caliciviridae*	Norovirus GI	RT-qPCR	QNIF4: CGCTGGATGCGNTTCCAT NV1LCR: CCTTAGACGCCATCATCATTTAC NVGG1p: FAM- TGGACAGGAGAYCGCRATCT-BHQ1	[53,54]
	Norovirus GII	RT-qPCR	QNIF2: ATGTTCAGRTGGATGAGRTTCTCWGA COG2R: TCGACGCCATCTTCATTCACA QNIFS: FAM- AGC ACG TGG GAG GGC GAT CG -BHQ1	[55,56]
*Reoviridae*	Group A rotavirus	RT-qPCR	JVKF: CAGTGGTTGATGCTCAAGATGGA JVKR: TCATTGTAATCATATTGAATACCCA JVKP: FAM-ACAACTGCAGCTTCAAAAGAAGWGT-BHQ	[57]
*Arenaviridae*	Mammarenavirus	RT-PCR, nested	Arena-F1: AYNGGNACNCCRTTNGC Arena-R1: TCHTAYAARGARCARGTDGGDGG Arena-F2: GGNACYTCHTCHCCCCANAC Arena-R2: AGYAARTGGGGNCCNAYKATG	[58]
*Picornaviridae*	Kobuvirus	RT-PCR	UNIV-kobu-F: TGGAYTACAAG(/R)TGTTTTGATGC UNIV-kobu-R: ATGTTGTTRATGATGGTGTTGA	[59]
*Astroviridae*	Mamastrovirus	RT-PCR, nested	Mon269: CAACTCAGGAAACAGGGTGT Mon270: TCAGATGCATTGTCATTGGT Mon269N GACCAAAACCTGCAATATGTCA	[60,61]
		RT-PCR	SF0073: ATTGGACTCGATTTGATGG SF0076: CTGGCTTAACCCACATTCC	[62]

**Table 2 viruses-16-01041-t002:** Description of the characteristics of examined animals.

Specie	Sex	Age Class (%)		Weght (g)			Samples			
		Adult	Subadult	Min	Max	Median	Fecal	Lung	Liver	Total
*Mus musculus* (House mice) (n = 13; F = 2, M = 11)	F	2 (100)	0 (0.0)	17	20.4	18.7	2	2	2	6
	M	10 (90.9)	1 (9.1)	7	70	22.1	6	9	4	19
*Rattus norvegicus* (Brown rats) (n = 70; F = 24, M = 46)	F	18 (75.0)	6 (25.0)	45	390	195.2	22	23	20	65
	M	22 (47.8)	24 (52.2)	28.4	470	156.5	41	36	34	111
*Rattus rattus* (Black rats) (n = 45; F = 17, M = 28)	F	12 (70.6)	5 (29.4)	29	195	97.6	16	15	14	45
	M	15 (53.6)	13 (46.4)	18	165	87.6	24	18	11	53
Total (n = 128)		79 (61.7)	49 (38.3)	-	-	-	111	103	85	299

**Table 3 viruses-16-01041-t003:** Number of animals studied and types of samples analyzed per animal. For each sample type, the detected viruses and the number of positive samples are provided.

Specie *	Sample Type	No. of Samples Tested	Detected Viruses (No. of Positive Samples)	
*Rattus norvegicus* (Brown rats)	Fecal/rectal swab	63	MukV	(6)
			AstV	(1)
	Liver	54	-	(0)
	Lung	59	MukV	(1)
*Rattus rattus* (Black rats)	Fecal/rectal swab	40	MukV	(7)
	Liver	25	-	(0)
	Lung	33	-	(0)

* The results for *Mus musculus* were not reported since all samples tested negative.

## Data Availability

Data is contained within the article.

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
