# Peer review of "Exploring the Potential of Muridae as Sentinels for Human and Zoonotic Viruses"

_viruses, 2024, doi:10.3390/v16071041_

Round 1

Reviewer 1 Report

Comments and Suggestions for Authors

I have no suggestions for the authors other than to review the English and the style of the text.

After checking the English style in the entire text I recommend the publication of the manuscript

Comments on the Quality of English Language

a moderate review of English quality is required

Author Response

I have no suggestions for the authors other than to review the English and the style of the text.

After checking the English style in the entire text I recommend the publication of the manuscript.

Answer: The text has been carefully revised to improve the quality of the English language.

Reviewer 2 Report

Comments and Suggestions for Authors

Human activities are increasing the risk of zoonotic diseases originating from wild animals. In this study, the authors investigated the prevalence of pathogenic viruses in small rodents, mice and rats, which are urban wild animals. These animals are reservoirs for various pathogens that cause severe infections in humans. As a result of this study, RNA sequences with high sequence similarity to both human and rat astroviruses were found in fecal samples of Rattus norvegicus. In addition, mouse kobuvirus RNA was detected mainly in fecal samples of Rattus rattus and Rattus norvegicus from urban areas. These results revealed that pathogens from urban rodents potentially carry viruses pathogenic to humans, and the authors argue that further monitoring is necessary. I think this is an important study for public health.

[Major points]

The contents of the paper is simple and easy to understand, and I think it is an important research topic. However, if it is to be a "full-paper", it would be better if the following figures and tables were included.

1. A map showing the sampling locations and the characteristics of their natural environment

2. A table showing the more detailed characteristics of the sample mice and rats, such as sex and weight

3. Phylogenetic tree analysis based on the obtained virus-derived base sequences

4. Etc.

Or, I think it would be better to change it’ format to "short communication".

[Minor points]

1. Lines 177-190: Wouldn't it be better to make "3.1. Characteristics of examined animals" into a table?

2. Lines 230 and 306: Isn't "species-specific" wrong instead of "species-specific"?

Author Response

Human activities are increasing the risk of zoonotic diseases originating from wild animals. In this study, the authors investigated the prevalence of pathogenic viruses in small rodents, mice and rats, which are urban wild animals. These animals are reservoirs for various pathogens that cause severe infections in humans. As a result of this study, RNA sequences with high sequence similarity to both human and rat astroviruses were found in fecal samples of Rattus norvegicus. In addition, mouse kobuvirus RNA was detected mainly in fecal samples of Rattus rattus and Rattus norvegicus from urban areas. These results revealed that pathogens from urban rodents potentially carry viruses pathogenic to humans, and the authors argue that further monitoring is necessary. I think this is an important study for public health.

Answer: Dear Reviewer, on behalf of all the authors, I would like to express our sincere gratitude to you for appreciating our manuscript.

[Major points]

The contents of the paper is simple and easy to understand, and I think it is an important research topic. However, if it is to be a "full-paper", it would be better if the following figures and tables were included.

  1. A map showing the sampling locations and the characteristics of their natural environment.

Answer: As suggested by the reviewer, a map was added (Figures 1) where the municipalities from which the mice and rats analyzed came are highlighted (line: 176).

  1. A table showing the more detailed characteristics of the sample mice and rats, such as sex and weight

Answer: We agree with the reviewer's comment. A new table (Table 2) shows in detail the characteristics of the animals examined (line: 247).

  1. Phylogenetic tree analysis based on the obtained virus-derived base sequences

Answer: we excluded phylogenetic analyses for the limited number of sequences available, since for kobuvirus as reported in the text, all sequences were closely related, and only one astrovirus was positive and sequenced. The information about nucleotide sequences identity is already present in the text. Moreover, the length of sequences was too short for a robust analysis. We thanks to reviewer for the interesting comment which give us a suggestion for future work on longer sequence but in this paper we prefer not to include it.

  1. Etc.

Or, I think it would be better to change it’ format to "short communication".

Answer: We had chosen a full paper format to allow a better framing of the topic regarding the potential role of Muridae in the epidemiology of viral zoonoses. In agreement with the suggestions provided by the Reviewer (see our previous answers), additional information was added to allow the format of the paper not to be changed.

[Minor points]

  1. Lines 177-190: Wouldn't it be better to make "3.1. Characteristics of examined animals" into a table?

Answer: We agree with the reviewer's comment. A new table (Table 2) shows in detail the characteristics of the animals examined. Accordingly, paragraph 3.1 has been modified.

  1. Lines 230 and 306: Isn't "species-specific" wrong instead of "species-specific"?

Answer: Thanks to the reviewer for pointing out the typo.

Reviewer 3 Report

Comments and Suggestions for Authors

I want to congratulate Di Bartolo and co-authors for an interesting study. Zoonoses are diseases transmitted from vertebrate animals to humans and are considered one of the most important threats to Public Health. Rodents can act as reservoir hosts of many zoonotic viruses and are known to play an important role in their transmission. The manuscript “Exploring the potential of Muridae as sentinels for human and zoonotic viruses” is well written and interesting to read.

Major comments:

My only negative comment is that you did not include the Hepatitis E virus in the study. Rats are a possible natural reservoir for HEV because they interact with people and domestic animals closely and regularly.  My recommendation is to design a new study and test the rat liver samples for the presence of HEV RNA. In 2022, the first case of Orthohepevirus C in acute HEV-infected patients in Europe was reported in Spain. Rat hepatitis E in humans has also been reported in France. Therefore, it is important to monitoring the prevalence of HEV among the rat population.

These recommendations do not detract from the quality of the study, and I believe that this manuscript should be published in Viruses.

Minor comments:

L 50 there is limited information available regarding the viruses carried by rats or mice that may pose a zoonotic threat to human health. Please rewrite the sentence, rats and mice are one of the most studied animals for the presence of zoonoses.

L56-57 Please, give more information about Orthohantaviruses and there distribution in Europe.

I would like to see specific examples of rodent-borne zoonoses spread in Europe to show why you chose to study these viruses in particular

In the introduction, it might be good to mention the Hepatitis E virus and the role of rodents in its spread.

Argue why you excluded the Juvenile animals from the study.

L243-255 - The beginning of your discussion is more like an introduction, please rewrite it and move the information into the introduction.  

Author Response

I want to congratulate Di Bartolo and co-authors for an interesting study. Zoonoses are diseases transmitted from vertebrate animals to humans and are considered one of the most important threats to Public Health. Rodents can act as reservoir hosts of many zoonotic viruses and are known to play an important role in their transmission. The manuscript “Exploring the potential of Muridae as sentinels for human and zoonotic viruses” is well written and interesting to read.

Answer: Dear Reviewer, on behalf of all the authors, I would like to express our sincere gratitude to you for appreciating our manuscript.

Major comments:

My only negative comment is that you did not include the Hepatitis E virus in the study. Rats are a possible natural reservoir for HEV because they interact with people and domestic animals closely and regularly.  My recommendation is to design a new study and test the rat liver samples for the presence of HEV RNA. In 2022, the first case of Orthohepevirus C in acute HEV-infected patients in Europe was reported in Spain. Rat hepatitis E in humans has also been reported in France. Therefore, it is important to monitoring the prevalence of HEV among the rat population.

These recommendations do not detract from the quality of the study, and I believe that this manuscript should be published in Viruses.

Answer: Thank you for the comment, we agree and a sentence in the introduction was added to complete the information on viruses circulating in rats (lines: 71-78). We have already planned a paper on HEV-C in rats, which has been submitted (under review) and was focused on phylogenies. In the present study, we investigated viruses not commonly detected before in Italy.

Minor comments:

L 50 there is limited information available regarding the viruses carried by rats or mice that may pose a zoonotic threat to human health. Please rewrite the sentence, rats and mice are one of the most studied animals for the presence of zoonoses.

Answer: We agree with the Reviewer's comments. The sentence has been rewritten (lines: 46-48).

L56-57 Please, give more information about Orthohantaviruses and there distribution in Europe.

Answer: A text (lines: 52-54, 83-89) describing Orthohantaviruses and there distribution in Europe was added in introduction and some information about this topic reported in the discussion section were moved to the introduction.

I would like to see specific examples of rodent-borne zoonoses spread in Europe to show why you chose to study these viruses in particular

Answer: The reviewer is correct, and we have clarified some information yet present in the introduction and discussion sections (Hantavirus: lines 80-99, 334-339; Arenavirus lines: 104-108, 345-347; MPXV lines: 110-117, 351-355;  CoVs lines: 121-125, 364-366; Sars-Cov2 lines: 123-124, 360-363; Astrovirus lines: 155-158, 371-376; Norovirus lines: 150-152, 397-403; Rotavirus lines: 141-144, 406-413; Kobuvirus lines: 129-134, 414-421).

In the introduction, it might be good to mention the Hepatitis E virus and the role of rodents in its spread.

Answer: We agree with the reviewer, some information about Hepatitis E virus and the role of rodents in its spread was added in introduction (lines: 51-54, 74-76).

Argue why you excluded the Juvenile animals from the study.

Answer: Juvenile animals were not excluded a priori from the study. Mechanical traps were used to capture the animals. This type of trap is designed to allow the capture of adult/subadult animals. The juvenile category is able to escape from this type of traps.

L243-255 - The beginning of your discussion is more like an introduction, please rewrite it and move the information into the introduction. 

Answer: We agree with the reviewer, this section was modified and moved into the introduction.

Round 2

Reviewer 2 Report

Comments and Suggestions for Authors

I think it's much better than the original version. The authors' intentions are more clearly expressed by showing figure and tables. I have no further comments.